# Hubbing the Cancer Cell

**DOI:** 10.3390/cancers14235924

**Published:** 2022-11-30

**Authors:** Jingkai Zhou, Matthieu Corvaisier, Darina Malycheva, Maria Alvarado-Kristensson

**Affiliations:** Molecular Pathology, Department of Translational Medicine, Skåne University Hospital Malmö 1, Lund University, 20502 Malmö, Sweden

**Keywords:** actin, microtubules, γ-tubulin meshwork

## Abstract

**Simple Summary:**

Cancer originates from changes in the genetics of single cells that affect their proliferative rate. To cope with the increased demand of building blocks and energy, tumor cells undergo adaptive changes creating new cellular homeostasis. These newly acquired traits are used clinically as diagnostic markers. Here, we summarize our knowledge of how a cell can adjust to new energetic demands during the transformation into a tumor cell.

**Abstract:**

Oncogenic transformation drives adaptive changes in a growing tumor that affect the cellular organization of cancerous cells, resulting in the loss of specialized cellular functions in the polarized compartmentalization of cells. The resulting altered metabolic and morphological patterns are used clinically as diagnostic markers. This review recapitulates the known functions of actin, microtubules and the γ-tubulin meshwork in orchestrating cell metabolism and functional cellular asymmetry.

## 1. Introduction

The glucose craving of cancer cells is used clinically to visualize tumors via positron emission tomography (PET) scanning. Patients are injected with a nonmetabolizable labeled glucose derivate that is actively consumed by dividing tissues, and hence, rapidly growing tumors are detected [1]. This high level of glucose uptake, the Warburg effect [2], is also observed in rapidly growing tissues, such as embryonic tissues [3], and reflects an increased anaerobic metabolization of glucose into pyruvate and, ultimately, lactate through glycolysis. The anaerobic metabolization of glucose occurs in the cytosol and generates two molecules of adenosine triphosphate (ATP) per glucose molecule, in contrast to the mitochondrial oxidative phosphorylation, where cytosolic pyruvate enters the mitochondria and becomes a substrate in the citric acid cycle, generating 36 molecules of ATP per glucose molecule. Glycolysis is the preferable metabolic pathway used by cancer cells, as the pathway produces metabolites required for the production of cellular components, which are a prerequisite for the execution of cell proliferation despite a lower yield of ATP [4,5].

The end product of the anaerobic metabolization of glucose, lactate, is also a glucose precursor in gluconeogenesis. Lactate can convert back into pyruvate and enter the mitochondria, where the first steps of gluconeogenesis occur [6]. Glycolysis, oxidative phosphorylation, and gluconeogenesis are good examples of the crosstalk between the cytosol and mitochondria, which is indispensable for the metabolism of tumor cells. Part of the biochemical reactions in cell metabolism is enzyme-catalyzed by genetically encoded proteins in the mitochondrial genome, but most of the enzymes required for mitochondrial function and biogenesis are encoded by nuclear genes [7]. To solve this cellular arrangement, the cells have evolved communication paths between compartments. One such path is the reciprocal regulatory signaling network established between the mitochondria and the nucleus [7].

The compartmentalization of eukaryotic cells generates a functional asymmetry with varying energy and macromolecule demands. This is especially important for the maintenance of specialized cellular functions that rely on cell polarization. Thus, orchestrating cell metabolism requires structural scaffolds that facilitate transport and create a venue for both polarization and biochemical reactions. Two important aspects of tumor progression, invasion and proliferation, are promoted by cytoskeletal and metabolic rearrangements [8]. In-depth knowledge of the processes underlying tumor proliferation may refine clinical applications. Here, we review and discuss the latest advances in the field regarding interactions between organelles (with a focus on the mitochondria and nucleus) and cytoskeletal elements (actin, microtubules, and the γ-tubulin meshwork) that orchestrate and nurse the cell proliferation and cellular metabolism of healthy cells and under pathological conditions.

## 2. The Need for Communication between Compartments

Cellular proliferation requires environmental changes to be continuously sensed, processed and responded. The cell membrane receives and transmits environmental information to the cell organelles, which adapt and respond to the signal. Plasma membrane receptors transduce extracellular information by generating secondary messengers, such as phosphorylation of cytosolic proteins, inositide phosphates, calcium or cyclic adenoside monophosphate, that alter the interaction between proteins and lipids [9]. The opening and closing of ion channel receptors can alter the flux of ions across the plasma membrane, altogether triggering a signal-transducing chain of events in the cytoplasm that induce cytoskeletal rearrangements [10]. Moreover, receptor binding can lead to the uptake and internalization of environmental molecules through endocytosis, generating cytosolic vesicles that traffic cells to acceptor organelles located in various cellular regions (Figure 1). Vesicle trafficking is based on the fusion and budding of vesicles and requires the structural support of various cytoskeletons [11].

### 2.1. The Interactions between Mitochondria and the Nucleus

Each cellular organelle is defined by the metabolic processes that are carried out in that location, and for the accomplishment of the processes, there is a further spatial compartmentalization within organelles. One example is the mitochondria which structurally consist of various membranes: an outer membrane, an intermembrane space, an inner membrane, and a matrix. To some extent, small cytosolic molecules can freely diffuse into the intermembrane space through porins located in the mitochondrial outer membrane (MOM), whereas protein transporters manage the import of molecules through the impermeable inner membrane. The respiratory complexes in the electron transport chain, which convert substrates into ATP, are located in membrane protrusions of the mitochondrial inner membrane (MIM), known as cristae. The MIM and MOM meet at contact sites that are involved in cristae organization [12]. The matrix is the innermost mitochondrial space, where the many copies of the double-stranded and circular mitochondrial (mt) deoxyribonucleic acid (DNA) are found, and there, processes such as mtDNA replication and transcription, the tricarboxylic acid (TCA) cycle and protein synthesis take place. In the matrix side of the MIM, the mtDNA and associated proteins form nucleoids. Mammalian mtDNA encodes for respiratory protein complexes and for mitochondrial ribosomal and transfers ribonucleic acids (RNAs).

Similar to the mitochondria, the nucleus has an inner (NIM) and an outer nuclear membrane (NOM) that are joined by nuclear pores complexes (NPCs) and enclose the histone and non-histone protein compacted DNA (chromatin). Molecule trafficking into and out of the nuclear compartment is controlled by NPCs and is necessary for the replication and transcription of the chromatin [13]. Another similarity with the mitochondria was highlighted in a recent study. Kafkia and colleagues demonstrated in human HeLa cells and murine embryonic stem cells that part of the metabolic reactions in the TCA cycle can also occur in the nucleus and that the resulting metabolites are critical for the modification of RNA and chromatin at that site [14].

Maintenance of the spatial organization of organelles demands the supply of substrates from other cellular locations for the running of biochemical reactions. To cover for these needs, there is transient contact or apposition between membranes that create ways of inter-organelle communication among organelles [15]. For instance, regions of the endoplasmic reticulum (ER), a continuous system of membranes that enclose a common luminal space, are in contact with mitochondria [16]. These contact sites are important for the coordination of cellular functions, such as calcium signaling and lipid transfer, and determine the sites for mitochondrial division [15,17]. Other membrane contacts between endosomes and mitochondria supply iron to mitochondria [18]. There are also points of contact between mitochondria and the nucleus, which favor the retro-communication of the mitochondria with the nuclear compartment (Figure 1) [19]. Nonetheless, the ER is contiguous with the NOM, and thus, communication between the nucleus and the mitochondria can also occur through the ER [20]. 

### 2.2. Cellular Skeleton

Eukaryotic cells have evolved proteins that can self-assemble into three-dimensional fibers and/or hollow rods, creating cytoskeletal/nucleoskeletal networks that serve as upholders of the cellular structure. The length of the fibers can dynamically change, and the fibers can be used as scaffolds for the docking of metabolic enzymes, intracellular movements, vesicular trafficking and cell division. The arrays of the various networks, i.e., actin, microtubules, the γ-tubulin meshwork and intermediated filaments, extend from the nuclear envelope to the plasma membrane and maintain the position and structure of cellular organelles [21,22]. 

#### 2.2.1. Actin

Defects in nuclear shape are a common trait of cancer cells and are clinically used as a diagnostic marker. An abnormal shape can be partially explained by a modified actin cytoskeleton, which is built of actin monomers that can polymerize into linear polarized filaments (F-actin) in an ATP-dependent manner (Table 1). In adherent cells, cytosolic actin together with the motor protein myosin II forms bundles of contractile actomyosin fibers, named the actin cap (Figure 1). This perinuclear actin cap is connected apically to the nuclear envelope and lamina through linkers of nucleoskeleton and cytoskeleton (LINC) complexes and basically associates with a subset of focal adhesions in the plasma membrane at the cell edges, connecting the cell to the extracellular matrix [21,22]. These contractile actomyosin fibers typically run along the axis of a cell and can generate contractile forces upon external stimuli. In adherent cells, the actin cap maintains a thin and discoidal nuclear shape in interphase nuclei, regulates the translocation and rotation of the nucleus during shear response and migration, and plays a role in sensing changes in extracellular matrix compliance and mechanical forces [21,22]. The actin cap is a direct connection between the plasma membrane and the nucleus, and it is proposed to be a dynamic network that allows a cell to rapidly respond to an extracellular signal with a genomic rearrangement (Table 1). The actin cap has been shown to be disassembled in various cancer cell lines [21,23].

In addition to connecting the nuclear envelope with the plasma membrane, actin filaments form a contractile ring of actomyosin fibers that physically separate offspring cells during mitosis, merging the plasma membrane of each resulting cell [24]. In most eukaryotic cells close to the plasma membrane, cortical actin establishes a meshwork that contributes to the distribution and motility of membrane junctions formed between the ER and the plasma membrane [25,26]. The ER-PM junctions aid in non-vesicular lipid transfer and in the intracellular regulation of calcium signaling [27,28]. Additionally, actin can produce forces that reshape and facilitate plasma membrane trafficking, assisting in the uptake of environmental molecules through the formation of cytosolic vesicles (Figure 1) [29]. Once formed, intracellular vesicles are transported on actin fibers (Table 1) [30]. 

The morphology of cellular mitochondria is the result of a fission and fusion balance that is regulated by dynamin-related protein 1 (Drp1, mediates mitochondrial fission), mitofusin 1 (Mfn1) and optic atrophy 1 (OPA1, both mediate mitochondrial fusion) [31]. In HeLa cells, cycles of actin polymerization/depolymerization affect the fission/fusion dynamics of mitochondria, changing the mitochondrial morphology [32]. During early cytokinesis, mitochondria localize to the contractile actomyosin ring, assuring an even distribution between offspring cells [33]. Actin is also described to establish plasma-membrane–mitochondrial bridges at the apical membrane of astrocytes and T-cells [34], as well as mediating mitochondrial transport and anchoring at other cellular sites [35,36,37], altogether affecting both the morphology and the location of mitochondria (Table 1). The location of mitochondria serves the environment with calcium buffering and ATP production [38]. Thus, anchoring these organelles to the correct cellular site is important for specialized cellular functions [38]. Similarly, the enzyme creatinine kinase (CK) is recruited to actin filaments for creating a cellular ATP-buffering system through the reversible catalysis of the formation of phosphocreatine (PCr) from creatin and ATP. The buffer works through the reversible phosphor transferring between ATP and creatin or between PCr and adenosine diphosphate, which occurs at ATP-producing and ATP-consuming sites, respectively [39]. Finally, activation of cytosolic glycolytic enzymes, such as aldolase or glyceraldehyde phosphate dehydrogenase, and exchange of metabolites across the mitochondrial outer membrane through voltage-dependent anion channels (VDACs) are controlled by direct binding to F-actin [40] and free monomers of actin, respectively [41,42].

Apart from its cytosolic location, actin is described to be a structural element in both the nucleus and mitochondria (Figure 1). Nuclear actin regulates chromatin and transcriptional events, which are important regulators of gene programs. Thus, the ratio between the nuclear and the cytoplasmic pools of actin affects cell differentiation [43,44,45,46,47]. Low concentrations of nuclear actin are shown to stall the activity of RNA polymerase II, leading to trimethylation of H3K27 and silenced expression at that location (Table 1) [48]. Similarly, mitochondrial actin regulates the expression of mitochondrial-encoded genes and supports mitochondrial-DNA replication [49]. The presence of actin in both the nucleus and mitochondria has been suggested to maintain the anterograde-retrograde mitochondrial, nuclear signaling [49,50].
Figure 1Actin, microtubules and the γ-tubulin meshwork hub cellular signals and organelles. 1. Actin filaments reshape the plasma membrane for the uptake of environmental molecules [29]. 2. Cytoskeletal bridges connect mitochondria to the plasma membrane [34]. 3. The transport of organelles, such as mitochondria, occurs on microtubules and/or actin filaments [35,36,37,51,52,53,54]. 4. Microtubules contribute to the formation of plasma membrane protrusions [55,56]. 5. The centrosome is a site that regulates the nucleation of actin and microtubules [57,58]. 6. γ-Tubulin ring complexes (γ-TuRC) nucleate microtubules and, together with pericentrin, form γ-tubules [59]. 7. A boundary form of γ-tubulin polymers (γ-strings) connects the nucleus to the cytoplasm [60,61]. 8. γ-Strings and actin are found inside the nucleus and mitochondria [49,50,61,62,63]. 9. In adherent cells, bundles of perinuclear actin and myosin form the actin cap, maintaining the form of the nucleus and connecting it with the plasma membrane [21,22]. 10. There are points of contact between mitochondria and the nucleus [19].
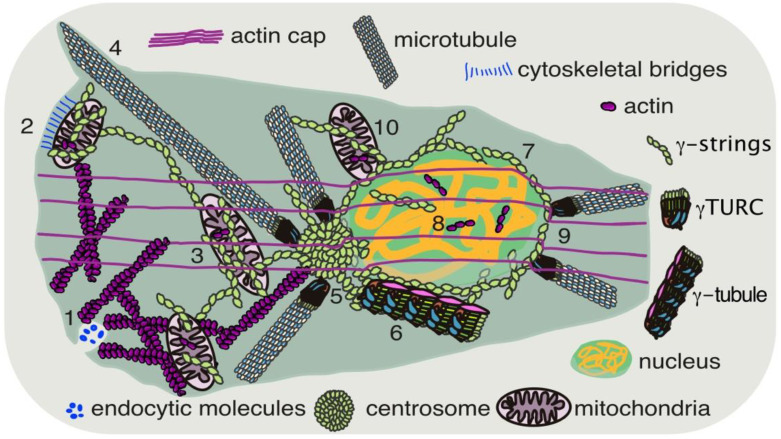

cancers-14-05924-t001_Table 1Table 1Comparison of the cellular functions of actin, microtubules and the γ-tubulin meshwork.FunctionReferencesMeshworkATPase[64]actinGTPase[65]microtubules, γ-tubulin ^1^Nuclear shape[21,22,66,67,68]actin, microtubules, γ-tubulin ^1^Nuclear formation[61]γ-tubulin ^1^Vesicle formation[29]actinVesicle trafficking[30,69]actin, microtubulesLocation of organelles[35,36,37,52,53,63,68]actin, microtubules, γ-tubulin ^1^Mitochondrial shape[49,63]actin, γ-tubulin ^1^Transcriptional regulation[43,44,46,48,63,70,71]actin, γ-tubulin ^1^DNA replication[70]γ-tubulin ^1^Nucleation activity[57,58,59,61]γ-tubulin ^1^Membrane component[61,63,70]γ-tubulin ^1^^1^ The γ-tubulin meshwork.


#### 2.2.2. Microtubules

Lagging chromosomes, aneuploidy and polyploidy are hallmarks of cancer cells and may be a consequence of a failed nuclear envelope rupture or flopped mitosis [72,73]. The nuclear envelope separates the chromosomes from the cytoplasm, and it is a continuous envelope during interphase. At the onset of mitosis, microtubule-mediated tearing of the nuclear envelope contributes to the initial nuclear permeabilization, which allows the microtubules in the mitotic spindle to obtain access to the condensed chromosomes [74]. Disassembly of the nuclear envelope entails the absorption of nuclear envelope membranes and associated proteins into the nuclear envelope-connected ER network (Table 1) [75,76]. Part of the dissembled pool of ER-nuclear envelope vesicles, as well as mitochondria, associates with spindle microtubules [77,78]. During nuclear formation, chromatin-bound proteins regulate microtubule dynamics, allowing the normal formation of the nucleus [79]. At the midbody, anti-parallel microtubule bundles perform various functions, from positioning the cleavage furrow to separating the resulting offspring cells [80,81,82]. In this context, a furrow microtubule array is a microtubule structure formed during cytokinesis that is required for the inclusion of membranes during furrow ingression [83].

Cytosolic microtubules also affect the morphology of the interphase nucleus and its movements (Table 1). An interplay between the nuclear lamina and microtubules affects the nuclear morphology, and mutations in the microtubule-associated protein tau are the underlying reason for the microtubule-mediated deformation of the neuronal nuclei of patients suffering from frontotemporal dementia [66,67]. The polarization of drosophila’s dorsal-ventral axis is a consequence of microtubules pushing the nucleus into oocytes. In this setting, microtubules nucleating on microtubule organization centers close to the nuclear envelope are the mechanical force that causes an invagination of the nucleus and its movement [68]. 

Interphase and mitotic microtubules provide polarized guiding tracks that are used by molecular motors for the intracellular transport of vesicles and organelles, such as mitochondria (Figure 1 and Table 1) [51,52,53,54,69]. Once at the targeted site, microtubules and actin can anchor mitochondria to, for example, the plasma membrane, forming plasma membrane–mitochondrial bridges in T cells (Table 1) [34]. Transport and anchoring are important cellular events for covering the energy-demanding regulation of the calcium concentration and local ATP levels at, for instance, presynaptic sites in neuronal axons [84]. In fission yeast (*Schizosaccharomyces pombe*), mitochondria are located along microtubule bundles, and these associations impede mitochondrial fission (Table 1) [85]. The local levels of ATP can be balanced through the association of CKs with microtubules in the mitotic spindle, as well as in the flagella of the sea urchin sperm [86,87]. Additionally, the components of microtubules, i.e., α-tubulin and β-tubulin, regulate the metabolite exchange across the mitochondrial outer membrane through the control of the permeability of the VDACs [41,88,89].

#### 2.2.3. The γ-Tubulin Meshwork 

Aneuploidy and tumorigenesis are connected to aberrant centrosome numbers and abnormal centrosomes [90,91,92,93,94]. Centrosomes are cellular organelles that participate in the maintenance of cell polarity, proliferation, and cell division [95]. They provide the mammalian cells with a site that regulates the nucleation of various cytoskeletal elements such as microtubules and actin (Figure 1) [57,58,96]. Centrosome movements guide the mitotic spindle and the exit from mitosis [74,97]. In each pole of the spindle, there is a centrosome that regulates the dynamics of microtubules, ensuring the strict segregation of offspring chromatids between the newly formed cells. The interphase centrosome is also constantly in motion around the nuclear envelope and assists in the formation of nuclear foci of proliferating cell nuclear antigen (PCNA) during DNA replication [70,98]. Movements of the nucleus and centrosomes are important during brain development in mice (Table 1) [99]. A genetic screen for mispositioned nuclei discovered an interaction of centrosomes with the components of the LINC complex, SUN (Sad1 and UNC-84) and KASH (Klarsicht, ANC-1, Syne homology), in model organisms [100,101,102]. SUN and KASH form a protein bridge in the nuclear envelope that connects the cytoskeleton to the nucleoskeleton [99,102]. The motor proteins dynein and kinesin, together with microtubules and the nucleoskeleton component lamin, maintain the position of a centrosome close to the nuclear envelope [103,104,105]. In a cell-free assay, actin and microtubules were shown to limit the position of centrosomes [106]. A notable fact is that a centrosome remains attached to the chromatin of demembranated Xenopus laevis sperm (depleted of membranes, microtubules, actin and lamina), suggesting a direct link between the chromatin and the centrosomes [61,98]. 

Centrosomes are rich in proteins with coiled-coil domains that facilitate protein–protein interactions, providing a scaffold onto which important cell-fate regulators can bind [107]. One such protein is the protein γ-tubulin, whose presence is necessary for the nucleation of microtubules onto centrosomes. Each γ-tubulin-containing complex, referred to as a γ-tubulin ring complex (γ-TuRC), creates a base for the growth of a microtubule (Figure 1) [58]. Nonetheless, cytosolic γ-TuRCs, together with the centrosomal protein pericentrin, can form 25 nm in diameter fibers called γ-tubules [59,108]. Beyond γ-TuRCs, centrosomal γ-tubulin in complex with pericentrin can build protein threads [109]. 

Similar threads of γ-tubulin (γ-strings) occur in the cytoplasm and are associated with cellular membranes and chromatin [59,61,63,71,110,111,112]. The chaperonin containing TCP-1 is involved in the correct cellular folding of γ-tubulin into γ-strings [113]. What is unique and special about γ-strings is their presence in the cytoplasm, membranes and chromatin (Figure 1 and Table 1) [61,62,63,70,110,111]. γ-Strings bridge centrosomes and γ-tubules to the nuclear compartment and chromatin, forming an interconnected meshwork that connects chromatin to the cytoplasm (Figure 1) [61,62]. 

A DNA-binding helix-loop-helix motif is enclosed in the C terminus of γ-tubulin (Table 1) [60,61]. Cells expressing a C-terminus-deleted γ-tubulin mutant were shown to form a nuclear envelope containing a lamina but to lack chromatin, implying that the C terminus is a prerequisite for connecting the formation of a nuclear envelope around chromatin [61]. Depending on the location of γ-tubulin, the appearance of the meshwork changes. At the G1–S transition, the activities of the transcription factor E2 promoter binding factor (E2F) trigger centrosome duplication and DNA replication. Concomitant with these events, the phosphorylation of γ-tubulin regulates the recruitment of the protein to the growing centrosome and chromatin. Once in chromatin, γ-tubulin turns off the transcriptional activities of E2Fs (Table 1) [114,115,116,117]. Chromatin immunoprecipitation sequence analysis and DNA fiber assay place γ-tubulin at the dormant origin of replications, and its presence is required for the loading of PCNA to chromatin during the S phase (Table 1) [70]. During early mitosis, the dispersed nuclear envelope components (with the exception of a γ-string boundary around chromatin) localize to the cell periphery and the mitotic spindle before reforming the nuclear envelope at the γ-tubulin boundary in late mitosis (Table 1) [61,76,118]. 

γ-Tubulin is also found to be associated with mitochondrial DNA. Mitochondrial-associated γ-strings provide mitochondria with a structural scaffold that links the mitochondrial DNA with the cytoplasm and functions as a regulator of the mitochondrial respiratory capacity (Figure 1 and Table 1) [63,119]. Impairment of the GTPase domain of γ-tubulin with the citral analog, citral dimethyl acetal, or with the metabolite fumarate (by treating cells with dimethyl fumarate [120]) was shown to impair the mitochondrial respiratory capacity, as well as depolymerizing γ-tubules, suggesting that the GTPase domain of γ-tubulin regulates the dynamics of the meshwork [59,63]. 

## 3. In Cancer Cells

The transition from differentiated to rapidly dividing undifferentiated tissue requires metabolic rewiring, as rapid proliferation demands increased building blocks for the assembly of new cells. To this end, the cancer cell increases the uptake of glucose for the biosynthesis of cell components rather than for the oxidative production of ATP [2,121]. Although this metabolic shift excludes the mitochondrial oxidative phosphorylation, the mitochondrial TCA cycle is required for the creation of intermediates that feed the synthesis of lipids, proteins and nucleic acids, making the mitochondria a metabolic biosynthesis hub in cancer cells [122,123,124]. Still, the total production of mitochondrial ATP in various tumor and non-transformed cells is equal since the cancer cell compensates for the lost glucose-dependent ATP production through an increase in the oxidation of carbon from other sources, such as glutamine and lipids [125]. 

In a growing tumor, oncogenic transformation drives adaptive metabolic changes that feed the increased energy demand of the spreading of tumor cells to distant organs [126]. During this process, the shape of mitochondria also adapts to the energetic requirements. There are several reports describing mitochondrial fragmentation as an adaptive metabolic strategy in tumor cells, where Drp1-dependent mitochondrial fission can promote increased glycolysis [127,128,129]. In line with this, Drp1-mediated mitochondrial fragmentation can be induced through the activation of Mitotic Activated Protein Kinase and oncogenic Ras signaling and is also observed in brain-tumor-initiating cells [128,129,130]. Another change in most tumors is the increase in expression levels of the mitochondrial CK1A that is needed for covering the local increased energetic demand of growing tumors [131].

The commitment to cell division is guarded by the tumor suppressor retinoblastoma (RB). The phosphorylation of RB triggers S-phase entry through the rupture of the RB–E2F complex, an event that activates the transcriptional activity of E2F [71]. Consequently, the acquirement of endless self-renewal ability requires the impairment of the function of RB, and thus *RB* is often found to be mutated in a large number of cancers [132]. In various tumors, low expression levels of RB coincide with an increased expression of γ-tubulin. Hence, the protein levels of γ-tubulin are increased in various tumors and coincide with high expression levels of both PCNA and E2F1 [70,71,117]. Furthermore, gene ontology enrichment analysis shows a positive correlation between the expression of *TUBG1* and cell-cycle-associated processes, suggesting that the expression of the γ-tubulin meshwork contributes to cell proliferation in tumor cells [70,71,117].

### Cancer Progression

Metastatic tumor cells can accumulate fragmented mitochondria at sites of actin remodeling at the leading edge for supplying energy during migration and invasion, and impairment of mitochondrial function impairs lamellipodia formation and migration of metastatic breast cancer cells [133]. In endometrial cancer, the high glucose environment induces mitochondrial damage that activates Drp1, resulting in increased proliferation, changes in glucose metabolism and epithelial–mesenchymal transition (a shift from epithelial cells to a mesenchymal phenotype) [134]. The gene expression profiling of metastatic circulating tumor cells showed a dysregulated oxidative phosphorylation and actin cytoskeleton [135]. Altogether, these studies reveal an intimate relationship between actin and mitochondria during migration and invasion. 

Microtubule tracks the delivery of metastatic components and molecules necessary for dynamic actin, as well as recycling adhesion molecules at the leading and/or rear edges of a migrating tumor cell (Table 1) [136,137]. Microtubules contribute to the generation of pushing forces at membrane protrusions and maintain polarity and directionally cell migration (Figure 1) [55,56]. In the migrating cell, microtubules move the nucleus to the cell rear with the centrosome creating a polarized axis in the direction of migration (Table 1) [138,139]. 

## 4. Conclusions

Cancer is a complex disease where a single cell can transform into a tumor cell that, through uncontrolled proliferation and, in some cases, metastatic invasion, can cause the death of a patient. Oncogenic transformation drives both adaptive metabolic and cytoskeletal changes that help the malignant cell to adapt to the proliferative and environmental demands [71,126,128,129,130,132]. This review recapitulates the known cellular functions of actin, microtubules and the γ-tubulin meshwork in metabolic, mitochondrial and nuclear homeostasis and considers their possible roles in cancer development. 

In an attempt to comprehend the various meshworks, we have listed their cellular functions in Table 1. In summary, their location differs; microtubules are cytosolic structures, whereas actin and the γ-tubulin meshwork are structures found in the cytoplasm, mitochondria and nucleus [48,49,61,63]. The coiled-coil domains in γ-tubulin facilitate protein–protein interactions with both microtubules and actin, suggesting that the γ-tubulin meshwork might constitute a link between actin and microtubules [57,58,96]. What also differentiates actin and microtubules from the γ-tubulin meshwork is that γ-tubulin has a DNA-binding domain, and it is also an integral component of both mitochondrial and nuclear membranes [60,61,63,70,71]. Moreover, actin and microtubules can participate in the transport of vesicles and affect the morphology and location of the nucleus and mitochondria, whereas actin and the γ-tubulin meshwork regulate DNA dynamics in both mitochondria and the nucleus [43,44,45,46,47,48,49,50,60,61,63,70,71]. Altogether, actin, microtubules and the γ-tubulin meshwork, along with their associated proteins, have distinct functions that cooperate to maintain the spatial organization of cells and hub the cell organelles.

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
