# Peer review of "Hubbing the Cancer Cell"

_cancers, 2022, doi:10.3390/cancers14235924_

Round 1

Author Response

Reviewer point #1

‘Hubbing’ is a neologism that meant ‘center or  pivot, core’…However, the nucleus genome is at the core of cell cancer changes, and the genome is not reviewed, such as DNAses and RNAses correcting defects in tumor cells – and this can be another topic –  It seems best that this head title word should be replaced by a more appropriate one.

Author response #1

Thank you for your comment. We agree with the meaning of the word and we think that we use it correctly. The aim of the review was to recapitulate how actin, microtubules and the gamma-tubulin meshwork are the core for the coordination of cell metabolism and functional cellular asymmetry. So for this reason, we would like to keep the word in the title.

Reviewer point #2

L75-76 that structurally consist of 75 various membranes: a…  should read ‘characterized’ or /and compartimentalized.

Author response #2

Thank you for your comment. In the new version of the manuscript we have changed the text accordingly with your suggestion.

Reviewer point #3

L109-111:  ‘communication between the nucleus and  the mitochondria may also occur through the ER’…please amend…  it occurred! The nucleus envelope is in continuity with the RER…it is part of the RER !  an example for it  [ Gilloteaux J,  Kashouty R,  Yono N: The perinuclear space of pancreatic acinar                            cells and the synthetic pathway of zymogen in Scorpaena scrofa L.:  Ultrastructural aspects.  Tissue & Cell 40: 7-20, and associated bibliography with Palade G who was involved in this consideration with biochemistry and cell fractionation and where the cited paper has showed with morphology].

Author response #3

Thank you for your comment. In the new version of the manuscript, we have changed the text accordingly to your comments.

Thank you for your time and consideration.

Reviewer 2 Report

The theme of the manuscript is not clear. The title is not framed well. Sections are not relatable. I could not understand the motive behind writing the manuscript.  Section 3 heading is "in cancer" What does it mean? This manuscript is not at all suitable for publication.

Author Response

Reviewer point #1

The theme of the manuscript is not clear. The title is not framed well. Sections are not relatable. I could not understand the motive behind writing the manuscript.  Section 3 heading is "in cancer" What does it mean? This manuscript is not at all suitable for publication.

Author response #1

Thank you for your comments. The aim of the review was to recapitulate how actin, microtubules and the gamma-tubulin meshwork are the core for the coordination of cell metabolism and functional cellular asymmetry. We also think that the title frammes well the content of the manuscript.

Thank you for your time and consideration.

Reviewer 3 Report

The authors in the review “Hubbing the Cancer Cell” have elaboratively described the process of oncogenic transformation. They have started with basic molecular processing including unusual glucose utilization in cancerous cells followed by cellular compartmentalization and communication in it. They have described the interaction between nucleus and mitochondria followed by cyto-skeletal macromolecules and how they are involved in multiple important processes like ATPase, GTPase, Nuclear shape, Vesicle formation and trafficking,  Mitochondrial shape as well as Transcriptional regulation providing a good understanding of the subject. 

This review is up to the mark for publication.

Author Response

Thank you for your comments, time and consideration.

Round 2

Reviewer 2 Report

I dont find this manuscript to be suitable for this journal.